# SNPs in Sheep: Characterization of Lithuanian Sheep Populations

**DOI:** 10.3390/ani11092651

**Published:** 2021-09-09

**Authors:** Ruta Sveistiene, Miika Tapio

**Affiliations:** 1Animal Science Institute, Lithuanian University of Health Sciences, 82317 Baisogala, Lithuania; 2Natural Resources Institute Finland, 00790 Helsinki, Finland; miika.tapio@luke.fi

**Keywords:** sheep, local breeds, genetic diversity, conservation

## Abstract

**Simple Summary:**

Nucleotide polymorphism (SNP) molecular markers were used to investigate the genetic variation in two native Lithuanian sheep breeds, compare them with the imported and theoretically close by origin Skudde sheep and establish the historical patterns of genetic relatedness to British, Central European and Nordic sheep breeds included in the SheepHapMap study. The results increase knowledge of and support for historical data on the importance of two local Lithuanian sheep breeds’ conservation and suggest the Skudde origin is less directly linked to the geographical area of modern-day Lithuania.

**Abstract:**

In Lithuania, there are two recognised native sheep breeds: old native Lithuanian Coarsewooled and Lithuanian Blackface. In addition, in 2005, primitive Heidschnucke-type Skudde sheep were imported to Lithuania and were argued to possibly represent a lost Lithuanian sheep type. The aim of the study was to investigate the genetic variation in the two Lithuanian native sheep breeds, compare them with the imported Skudde sheep and establish the historical patterns of admixture and the genetic relatedness of Lithuanian sheep to British, Central European and Nordic sheep breeds included in the SheepHapMap study. In total, 72 individuals, representing two Lithuanian native and imported Skudde sheep breeds, were genotyped using a Neogen 12K Illumina Infinium chip. The population analysis was carried out by model-based clustering, principal component analysis and neighbour net analysis, and showed similar patterns for the Lithuanian sheep populations. Lithuanian Coarsewooled and Skudde in Lithuania have unique divergence and possibly some shared ancestry, while the Lithuanian Blackface conforms to a modern synthetic breed. The study clearly showed that the Coarsewooled and the Skudde breeds are distinct from each other. Historical data strongly suggest that the Coarsewooled breed represents a local breed, while the Skudde origin is less directly linked to the geographical area of modern-day Lithuania. Within the modern-day Lithuanian context, the Lithuanian Coarsewooled sheep is very important historical sheep type for conservation.

## 1. Introduction

In Lithuania, there are two recognised native sheep breeds: old native Lithuanian Coarsewooled and Lithuanian Blackface, developed in the 20th century. The imports of foreign breeds and purposeful native sheep improvement through crossing accelerated the decline of native sheep from the 19th century and influenced the creation of the new Lithuanian Blackface sheep breed in the 20th century [1]. In addition, in 2005, primitive Skudde sheep were imported to Lithuania. Their shared origin and uniqueness have been debated.

At the end of the 19th to the beginning of the 20th century, late maturing yet undemanding coarse-wooled sheep were mostly bred in Lithuania. There were two types of sheep: long-thin-tailed Pomeranians, mostly spread in southwest Lithuania, and short-tailed, of a northern type, mostly spread in the eastern districts of Lithuania. The differences between the two types of coarse-wooled sheep eventually became smaller due to crossbreeding, and sheep with the characteristics of both breeds became predominant. By the end of the 20th century, there were only single sheep with the features of local coarse-wooled sheep remaining [2]. Activities for the conservation of old Lithuanian native breeds were launched in 1994 and 1999 when a minimal herd of local coarse-wooled sheep was formed at the Institute of Animal Science (LIAS), preventing complete extinction of the breed [3]. In 2013, the number of purebred Lithuanian Coarsewooled sheep was 220, and the breed had the status of an endangered breed [4]. Currently, the population numbers 755 sheep entered in the studbook.

The Lithuanian Blackface sheep breed was developed in the middle of the 20th century by crossing indigenous coarse-wooled sheep with Shropshire and German Blackface rams. The breed was recognised in 1961 as a semi-fine-wooled sheep of the meat–wool type [1]. At present, it is the largest sheep population in Lithuania and the number of purebred Lithuanian Blackface sheep amounts to 8772, whereas in 2013, there were only 4226 sheep [4]. Lithuanian Blackface sheep are still popular among breeders and have an effective population size with a “no longer vulnerable” risk status. Selection programs for Lithuanian Coarsewooled and Lithuanian Blackface sheep have been officially approved [1].

In 2005, sheep of the Skudde breed were imported to Lithuania from Germany. Later, sheep of this breed were imported from Poland and Belgium. In 2011, the Lithuanian Skudde sheep breeders’ association was established. In 2013, there were 279 Skudde sheep recorded in the Lithuanian animal register; in 2020, the number amounted to 2697, but in the meantime, the breed has no officially approved stud book and selection program.

Skudde sheep are categorised as being of the Heidschnucke type, which is part of the Northern European short-tailed sheep group [5,6]. Heidschnucke-type sheep were common in Eastern Prussia and the Baltic region. Therefore, it might be possible that Skudde and Lithuanian coarse-wooled sheep breeds have the same origin. So far, we have not found any Lithuanian historiographical source in which the name of “Skudde” sheep would have been mentioned as a breed in our territory. After World War II, the Skudde population numbered about 200 sheep, found in the zoos of Berlin, Leipzig and Munich. Some sheep also reached West Germany together with the refugees leaving Eastern Europe. According to Sveistiene, [1] in 1989, 30 native non-purebred Lithuanian coarse-wooled sheep were exported to West Germany with the aim of extending the genealogical structure of the Skudde sheep breed in Germany. Skudde sheep have been restored as a result of long-term breeding work in both Eastern and Western Germany [5].

Two native Lithuanian breeds have been previously assessed using microsatellites, and the Lithuanian Blackface appeared to be very similar to the other modern Baltic sheep breeds, while the native Coarsewooled sheep were a separate population between the short-tailed and long-tailed types [7,8]. Nowadays, genomic characterisation is more commonly based on SNP markers, making this method superior to microsatellites for inferring population structure [9]. A recently developed genome-wide high-density ovine SNP array has provided a tool for investigating the genetic diversity at a high resolution, inferring the population history and mapping genomic regions. Imputation accuracy in sheep breeds can be improved using both high- and low-density SNP panels [10,11].

The aim of the study was to investigate the genetic variation in two native Lithuanian sheep breeds, compare them with the imported and theoretically close by origin Skudde sheep and establish the historical patterns of admixture and genetic relatedness of Lithuanian sheep breeds with British, Central European and Nordic sheep breeds.

## 2. Materials and Methods

### 2.1. Data Generation and Collection

In total, 72 individuals, representing the old native Lithuanian Coarsewooled sheep breed, the modern Lithuanian Blackface and the imported Skudde sheep breed (Figure 1), were blood-sampled with 24 samples each.

The samples from the two Lithuanian local sheep breeds were collected only from purebred animals entered in the studbook of the Lithuanian Sheep Breeders’ Association. The samples from Lithuanian Coarsewooled sheep were taken at the Institute of Animal Science (Baisogala), where the sheep population has been restored. The sheep selected for sampling were adequate for the breed genealogy structure, with the whole population of the sheep in Lithuania having 4 different lines and 4 families in the population, and using a strict selection programme. Almost all existing purebred individuals, including some progeny, were sampled. The samples from Lithuanian Blackface sheep were taken on the National Sheep farm “Šeduvos Avininkystė”. The sheep selected for sampling represented the genealogical structure of 12 families and 4 lines, and reflected the whole purebred sheep population in Lithuania. Skudde sheep individuals from different stocks were sampled using limited pedigree information (mostly only parents and offspring) available in the unofficial register of the Lithuanian Skudde Sheep Breeders’ Association, and some sheep were sampled on the basis of import certificates indicating the sheep’s origin. We selected sheep imported from Poland, Germany and Belgium, and their progeny born in Lithuania. Blood samples were collected in tubes containing Ethylenediamine tetraacetic acid (EDTA). DNA was isolated from the blood samples. The total DNA was extracted using a genomic DNA purification kit (K0512; Fermentas, Hanover, MD, USA) according to the manufacturer’s instructions.

### 2.2. SNP Genotyping

In total, 72 individuals (24 sheep from each breed) were genotyped for 12,785 SNP markers. Additionally, the SNP data from British, Central European and Nordic sheep breeds were analysed. SNP genotyping was performed by the European headquarters of Neogen Corporation on an Illumina Infinium platform. The chip was a custom 12K chip. The genetic SNP analysis of the population was carried out by including only the mapped markers also available in the SheepHapMap SNP data [12]. Of this reference dataset, only Central and Northern European breeds relevant to the current study were included. The reference breeds used were Dorset Horn, Scottish Blackface, Wiltshire, Finnsheep, Black-Headed Mutton, East Friesian Brown, German Texel, Old Norwegian Spaelsau, Bundner Oberlander, Swiss Black-Brown Mountain, Swiss Mirror, Swiss White Alpine, and Valais Blacknose. The map information for both datasets was updated to the sheep genome (build v. 3.0) and only the autosome mapped markers present in both datasets were included. The SNP data quality control and further data management and dataset merging was performed using GenABEL (package v. 1.8-0) [13]. The markers with a minor allele frequency below 0.05 or a call rate below 98% were excluded. The redundant neighbouring markers were pruned out with plink v1.90a [14] using linkage disequilibrium-based sliding window pruning with a window size of 50 SNPs, a step size of 5 SNPs and a pairwise r^2^ limit of 0.1. After applying all filters, there were 9583 common SNP markers for the local and for the reference breeds available for statistical analyses. After inclusion of the Lithuanian samples, the SNP dataset had 509 sheep and 16 populations (Table 1).

### 2.3. Population Genetic Analyses

The population genetic analyses and data management were mainly carried out with *adegenet*, package v. 1.4.1 [15] within the *R* environment [16]. Variability was measured as expected and observed heterozygosity [17] and mean allele number. Deviations from Hardy–Weinberg genotype proportions were evaluated using the *HWE.test* function of *genetics*, package v. 1.3.8 [18]. Nei’s fixation indices were used to measure differentiation (*F_st_*) and inbreeding (*F_is_*) [17] using the *basic.stats* function in *hierfstat*, package v. 0.04–10 [19]. An in-house script, together with the *boot.ppfis* function in *hierfstat*, was used to statistically test the population-wise *F_is_* based on permutations and to estimate confidence intervals based on bootstrap samples.

The genetic structure was assessed using principal components analysis for scaled allele frequencies in *adegenet*. The *eucl.dist* function in *hierfstat* was used to calculate Euclidean distances between individuals based on their genotypes. The correlation between SNP-based distance matrices was tested with the Mantel test within the *R* environment. The ancestry of individuals was estimated using *Structure* v. 2.3 [20]. An admixture model where the allele frequencies may correlate and the markers are assumed to be unlinked was used. The number of assumed populations, k, varied from 1 to 18, with 3 independent repeats for each k using 30,000 burn-in iterations followed by 50,000 iterations. Phylogenetic relationships among populations were assessed on the basis of *NeighborNet* graphs, which were constructed from a matrix of pairwise *F_st_* estimates using *Splitstree* v. 4.13.1 [21].

## 3. Results

### 3.1. SNP Data

The mean call rate for Lithuanian samples with the custom 12K chip with 12,785 markers was nearly 100% across the markers and samples, with the lowest rates being 94%. The Illumina GenCall score (GC) varied from 0.81 to 0.98, suggesting very good technical quality. None of the markers deviated from Hardy–Weinberg proportions within the Lithuanian populations.

In total, 9583 mapped autosomal SNP markers were polymorphic and were included in the population analyses. The *F_st_* value based on the SNPs was 0.129. Though the standard deviations across the markers were approximately 0.1 among the sampled populations, this difference might not be stable. The marker set shows a clear structure and population relationships (Appendix A). The individual sheep were not greatly scattered around the population centre in the Principal Component Analyses (PCA) plot (Figure 2).

### 3.2. Lithuanian SNP Data with Reference Data

Lithuanian samples were analysed jointly with exotic breeds to reveal phylogenetic groupings and to compare genetic variability. Across the Lithuanian populations, the Lithuanian Coarsewooled had the smallest number of alleles (Table 1). The synthetic Lithuanian Blackface had the fourth largest number of alleles after Finnsheep, German Texel and Scottish Blackface. The Skudde population in Lithuania had the fifth smallest amount, similar to Dorset Horn. The other diversity measures correlated with this imperfectly. For example, with unbiased gene diversity, Wiltshire was less variable than the second least variable Lithuanian Coarsewooled, and Lithuanian Blackface became third most variable, this time after Old Norwegian Spaelsau and Scottish Blackface. In particular, the Lithuanian Coarsewooled breed, which has experienced a recent population size reduction, showed greatly elevated gene diversity compared with the allele number, while particularly East Friesian Brown and Dorset Horn appeared to have the opposite deviation, potentially related to population growth and mild admixture.

Differentiation in the full set was slightly stronger than within the Lithuanian set. The *F_st_* estimate for Central and Northern European breeds was 0.144.

Model-based clustering showed a mainly similar pattern to PCA. The three Lithuanian populations were resolved when the number of assumed populations, k, reached 14 (Figure 3). The Lithuanian Coarsewooled appeared as separate cluster already with k = 4, indicating the uniqueness of the population. The Skudde separated at k = 9. As also seen in the PCA analysis (Figure 4), there was one outlier sample in the Skudde. The clustering results suggest that the outlier might be a first-generation cross with a sheep type sharing ancestry with Bundner Oberlander and Black-Headed Mutton. The synthetic Lithuanian Blackface separates only at k = 14, and it appears to behighly admixed particularly when k = 6 to 13. Based on the model with k = 13, the breed had a significant (mean 5–25%) ancestry shared with Scottish Blackface, Wiltshire, German Texel, Lithuanian Coarsewooled, Valais Blacknose and Black-Headed Mutton, in order from largest to smallest. In the analysis, the model fit increased until k = 17 (Appendix A), but the results for the Lithuanian populations did not change (Appendix A).

The PCA showed that Lithuanian Coarsewooled and Skudde in Lithuania both have unique divergence and possibly some shared ancestry with exotic heritage breeds. The first four components in the PCA analysis (Appendix A) separated individual breeds (East Friesian Brown, Finnsheep, Lithuanian Coarsewooled and Dorset Horn), rather than grouping breeds together. Components 5 and 6 suggested the clustering of Lithuanian Coarsewooled with Wiltshire, and clustering of Skudde in Lithuania with Valais Blacknose Sheep, while the seventh component placed Skudde in Lithuania closer to Wiltshire. Component 8 separated Skudde in Lithuania as well as Valais Blacknose from each other and from the other breeds. Interestingly, the British Wiltshire sheep is a short-wooled meat breed known for the rare and primitive wool moulting trait. On the other hand, the Valais Blacknose is a hardy Swiss coarse-wooled breed with long wool.

The relationships among populations based on neighbour networks showed clear phylogenetic clusters of British, German, Swiss and Nordic breeds. The synthetic Lithuanian Blackface grouped between German and British breeds in the model based on clustering results. The two short-tailed populations in Lithuanian, Coarsewooled and Skudde, were placed around the Nordic cluster.

## 4. Discussion

The present study showed that two Lithuanian breeds and one imported sheep population each contain plenty of genetic variation and are very divergent from each other. Of the three studied Lithuanian sheep populations, the native Lithuanian Coarsewooled is the most unique, based on the genetic structure analyses presented. The native Lithuanian Coarsewooled was also the least variable, reflecting the narrow base of the breed restoration work, which started from six founders [4].

The results generally agreed with the earlier microsatellite results. However, the earlier microsatellite data indicated the native Lithuanian Coarsewooled breed grouping close to the modern panmictic populations [8], while the current SNP data analysis suggests a more distinct origin. Moreover, the within-breed variation in the current Coarsewooled sample does not equal that of the Lithuanian Blackface. In contrast, the results for the synthetic Lithuanian Blackface completely agreed with the earlier microsatellite results, as it was the most variable and showed its ancestry in British and German breeds. This is not surprising, as it is known from their historical origin that in 1963, the permanent introgression of geographically neighbouring breeds was used [1] and, according to Danta [22], Prussia Blackface sheep had the greatest effect.

Based on the SNP data, the imported Skudde sheep do not show similarity to Lithuanian breeds. We expected that Lithuanian Coarsewooled and Skudde would belong to the same breed group because of the historical and phenotype data [2,6,23]. This agrees with the clustering of populations based on microsatellites [7], where southern short- and semi-short-tailed breeds clustered between the northern short-tailed breeds and the long-tailed breeds. The Skudde breed clusters between the Nordic and German breeds. East Prussian Skudde sheep are mentioned as a transboundary breed bred in Germany, Switzerland and the Netherlands (http://dad.fao.org/, accessed on 6 January 2021), with several original forms across the countries [24].

Though the native Lithuanian Coarsewooled and Skudde do not group together, they both group between the Nordic and Central European breeds. According to our data, the variability in Skudde sheep was intermediate to the Lithuanian Coarsewooled and Lithuanian Blackface breeds. In general, the studied samples of the Skudde sheep in Lithuania were partly divergent from each other, most probably due to the influence of their import from different geographical regions. The clustering results suggest that the sheep might be a first-generation cross with a sheep type sharing ancestry with Bundner Oberlander and Black-Headed Mutton. Though this corresponds to the results of Ludwig Arne [25], who found genetic similarity between the Skudde and the Scottish breed Boreray, which he presumed to demonstrate the influence of Viking-age Scandinavian sheep on British flocks, the deduced crossing is very recent.

According to the SNP analysis, the Lithuanian Coarsewooled sheep breed is a small population with a narrow genetic diversity. Therefore, a conservation programme with an inbreeding control is important. When the native population is big enough (vulnerable status or better), similar to the like Lithuanian Blackface sheep created in the 20th century with quite large genetic diversity, selection for divergence can also be used. The generated information can be used for native Lithuanian sheep breeding and monitoring.

It is beneficial to look at the genetic data together with historical, geographical and social research. Historical data [1,21] strongly suggest that the Lithuanian Coarsewooled represents a local breed, while the origin of Skudde is less directly linked to the geographical area of modern-day Lithuania. The current results suggest that the Lithuanian Heidschnucke-type sheep’s contribution to Skudde might not have been significant. Therefore, within the modern-day Lithuanian context, the Lithuanian Coarsewooled sheep is the most important historical sheep type for conservation.

## 5. Conclusions

This is the first time that the population structure of the two native Lithuanian sheep breeds has been described using high-density SNP data. The native Lithuanian Coarsewooled sheep breed is generally a unique breed, and it is the most important historical sheep type for conservation. The Lithuanian Blackface sheep breed is more synthetic and groups between German and British breeds. The study clearly shows that the Lithuanian Coarsewooled and Skudde breeds are distinct from each other, and the historical data strongly suggest that the Lithuanian Coarsewooled represents a local breed, while the origin of Skudde is less directly linked to the geographical area of modern-day Lithuania.

## Figures and Tables

**Figure 1 animals-11-02651-f001:**
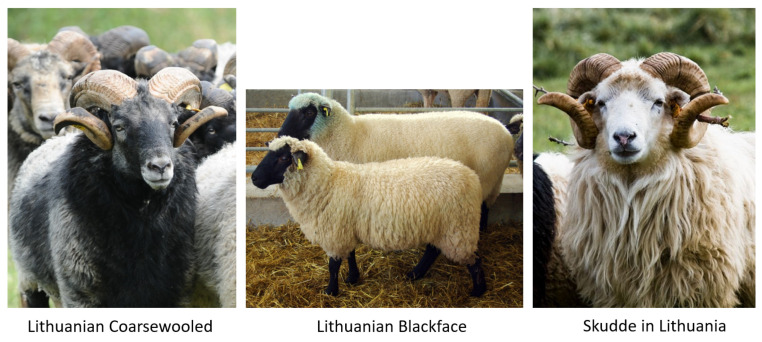
Pictures of two Lithuanian local sheep breeds and one imported Skudde sheep (Pictures made by A. Rackauskaite and K. Samusis).

**Figure 2 animals-11-02651-f002:**
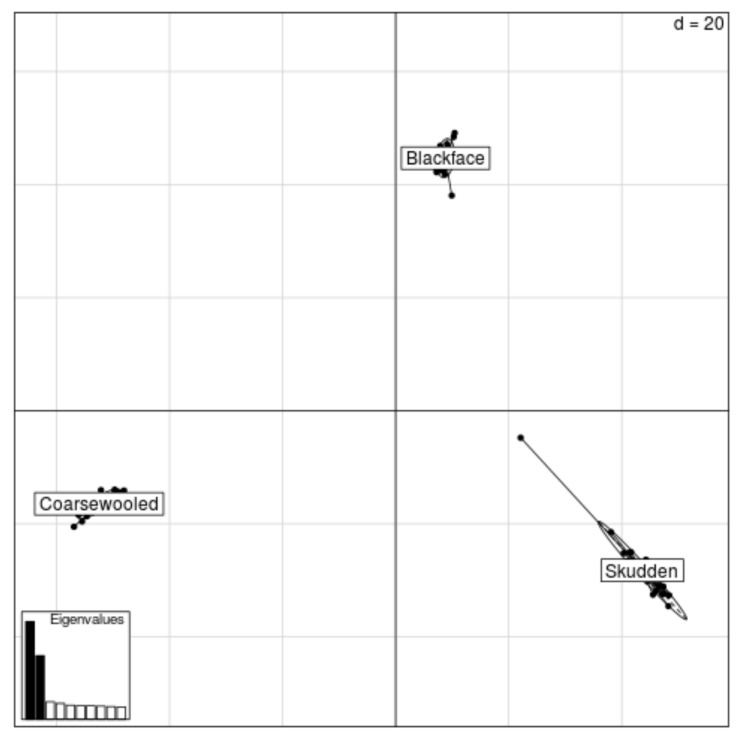
Principal component plots for the Lithuanian sheep, based on SNP data. As shown by the scree diagram, the two first components explaining 15% and 10% of the variation are sufficient for describing the main structure among the sheep, d-dimensional space.

**Figure 3 animals-11-02651-f003:**
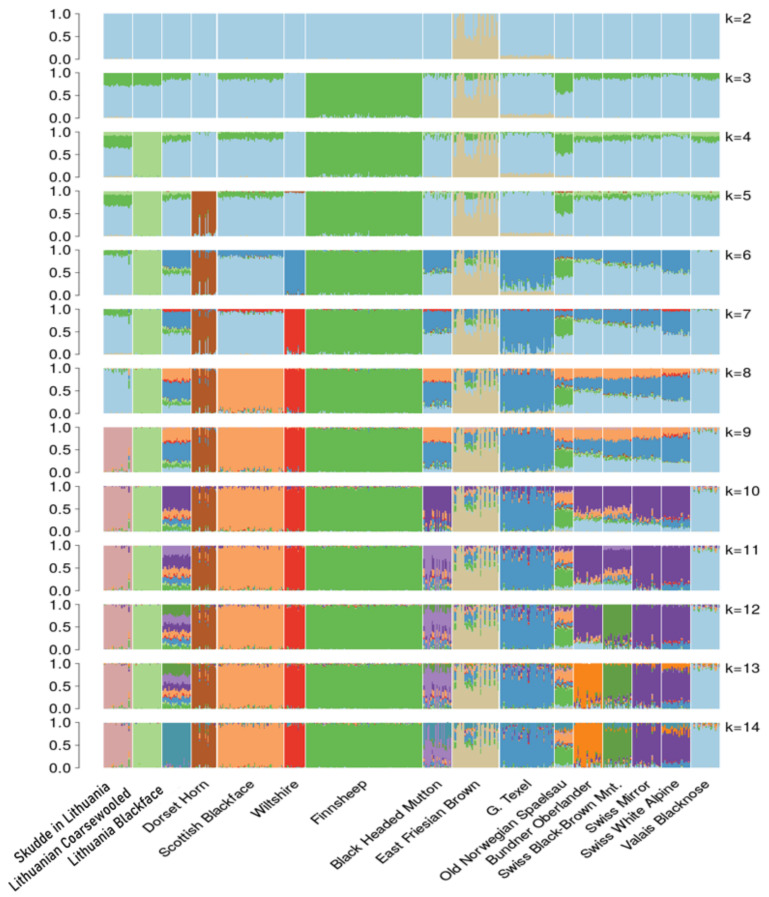
Bar charts describing the model-based clustering results for the models varying the assumed population number, k, from 2 to 14. Each thin bar represents an animal, and the bar is divided to different colours representing the estimated ancestry proportion for the individual animal. Sheep from the same population are grouped together and separated from the neighbouring breed by a space.

**Figure 4 animals-11-02651-f004:**
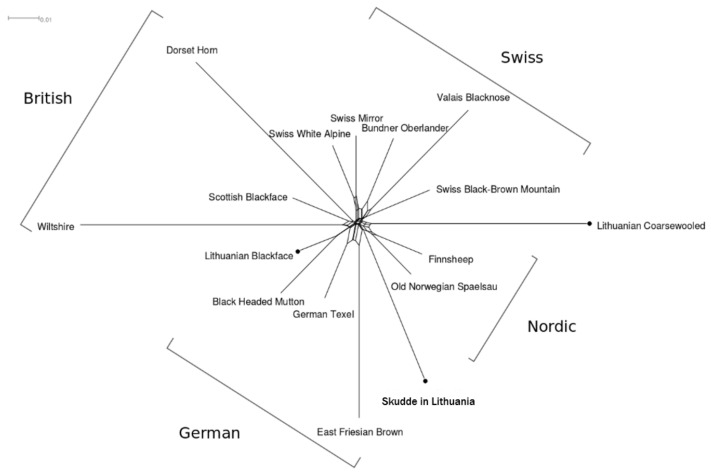
Neighbour-net network describing the relatedness structure among sheep breeds. The nodes for Lithuanian populations are emphasised with filled circles for clarity.

**Table 1 animals-11-02651-t001:** Studied sheep populations (Breed), the country of origin for the sample (Country), sample size (N), number of observed alleles (Nall), the proportion of marker polymorphism (Ppoly), mean minor allele frequency (MAF), observed heterozygosity (Hobs), unbiased gene diversity (Hexp), inbreeding coefficient (*Fis*) and number of significant (<0.0001) Hardy–Weinberg deviations per breed based on 9583 SNP markers.

Breed	Country	N	Nall	Ppoly	MAF	Hobs	Hexp	*Fis*	#HWEd
Lithuanian Coarsewooled	Lithuania	24	17,850	0.931	0.24	0.34	0.33	0.03	
Lithuanian Blackface	Lithuania	24	19,066	0.995	0.31	0.41	0.41	0.00	
Skudde in Lithuania	Lithuania	24	18,799	0.981	0.27	0.36	0.36	0.01	
Dorset Horn	Britain	21	18,799	0.981	0.25	0.36	0.34	0.06	
Scottish Blackface	Britain	56	19,108	0.997	0.32	0.41	0.41	0.00	
Wiltshire	Britain	17	18,173	0.948	0.22	0.31	0.31	0.01	1
Finnsheep	Finland	99	19,132	0.998	0.32	0.39	0.40	−0.03	2
Black-Headed Mutton	Germany	24	19,030	0.993	0.30	0.38	0.39	−0.03	
East Friesian Brown	Germany	39	18,891	0.986	0.26	0.34	0.34	−0.02	
German Texel	Germany	46	19,131	0.998	0.31	0.40	0.40	−0.02	3
Old Norwegian Spaelsau	Norway	15	19,006	0.992	0.31	0.38	0.41	−0.05	2
Bundner Oberlander	Switzerland	24	19,026	0.993	0.31	0.41	0.40	0.03	
Swiss Black-Brown Mountain	Switzerland	24	19,051	0.994	0.31	0.40	0.40	0.01	
Swiss Mirror	Switzerland	24	19,036	0.993	0.31	0.41	0.40	0.02	
Swiss White Alpine	Switzerland	24	19,064	0.995	0.31	0.41	0.40	0.01	1
Valais Blacknose	Switzerland	24	18,752	0.978	0.27	0.35	0.36	−0.05	
Mean		32	18,870	0.985	0.29	0.38	0.38	0.00	0.6

## Data Availability

The data presented in this study are available on request from the corresponding author.

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
