# Peer review of "SNPs in Sheep: Characterization of Lithuanian Sheep Populations"

_animals, 2021, doi:10.3390/ani11092651_

Round 1
Reviewer 1 Report
1)The manuscirpt perfomred the genentic divesity, genetic structure of Lithuanian s sheep populations using 12K SNP-chip, which provide useful information for sheep conservation and utilization in future.
2)The writing of the manuscript needs to be significanlty improved.
3)Generally, the SNPs numbers are reletively low, which a 12K-chip was used for genotyping. Besides allele numbers, other genetic diversity index should be compared in three local populations and other sheep breeds.
4)For the introduction section, the authors need introduce what recent research progress have been made on sheep diversity, population structure, selective signals, etc. And what methords has been applied for genomie- wide diversity(eg, gene chip ,resquencing )
5)It would be better that the authors provide the picture of three sheep breeds, which show the morphological characteristics of each populations.
Author Response
Authors thanks you for the comments and surggestions.
1) The manuscript performed the genetic diversity, genetic structure of Lithuanian s sheep populations using 12K SNP-chip, which provide useful information for sheep conservation and utilization in future.
Response: Thank you, for your supportive comment.
2) The writing of the manuscript needs to be significantly improved.
Response: we improved it
3) Generally, the SNPs numbers are relatively low, which a 12K-chip was used for genotyping. Besides allele numbers, other genetic diversity index should be compared in three local populations and other sheep breeds.
Response: The genetic diversity of three local populations is given in the Table 1.
4) For the introduction section, the authors need introduce what recent research progress have been made on sheep diversity, population structure, selective signals, etc. And what methods has been applied for genome- wide diversity (eg, gene chip, sequencing)
Response: Corrected, we add two new references
5) It would be better that the authors provide the picture of three sheep breeds, which show the morphological characteristics of each populations.
Response: We add the pictures of three analyzed populations in the Material and methods section.
Reviewer 2 Report
The authors compared two breeds of Lithuanian sheep with other European ships using SNP arrays. This work provides a framework for local sheep conservation.
Comments:
1) Please separate DNA collection and SNP genotyping in the section of Materials and Methods for clarity.
2) In page 4, line 161, what are populations used for estimating Fst? Same problem for Fst in page 5, line 185.
3) In Table 1, is “markers polymorphich” actually “marker polymorphism”?
4) It seems very unusual to me that L. Blackface showed less haplotypes when k=14 than k = 3~13, any reasonable explanation?
5) Should ‘Figure S2” in page 6 line 211 be “Figure S1”?
Author Response
Authors thanks for the comments and surggestions.
Comments:
1) Please separate DNA collection and SNP genotyping in the section of Materials and Methods for clarity.
Response: We have separated them for clarity
2) In page 4, line 161, what are populations used for estimating Fst? Same problem for Fst in page 5, line 185.
Response: Corrected – we add the populations.
3) In Table 1, is “markers polymorphich” actually “marker polymorphism”?
Response: corrected
4) It seems very unusual to me that L. Blackface showed less haplotypes when k=14 than k = 3~13, any reasonable explanation?
Response: The “haplotypes” in this case are several origins. The composite breed not very diverged and it is not clearly more similar to one other included breed. In this case, mixed ancestry looks as there.
5) Should ‘Figure S2” in page 6 line 211 be “Figure S1”?
Response: corrected
Reviewer 3 Report
Review report animals-1338741
The reviewed article entitled “NPs in sheep: Characterization of Lithuanian sheep populations” contributes to update the literature data about the genetic variation in two Lithuanian native sheep breeds and Authors compare them with the imported and “native” sheep breed- Skudde using new SNP data analysis. Authors have obtained novel and interesting results, which are presented in a simply and clear way. They Conclusions are appropriate and justified. However, I have minor suggestions in this work:
- For better understanding, Authors should standardize name of breed Skudde. We can see Skudde in the text, “Skudde in Lithuania” in Table and “L.Skudeen” in Figures
- sentences in 167-168 lines to 166 line (begin of the section)
- In line 224-The sentence “This agrees with the clusteing of populations based on microsatellites (7)” in my opinion should be placed in Discussion section.
Author Response
Authors thanks for the comments and surggestions.
1. For better understanding, Authors should standardize name of breed Skudde. We can see Skudde in the text, “Skudde in Lithuania” in Table and “L.Skudeen” in Figures
Response: corrected in the figures.
2. Sentences in 167-168 lines to 166 line (begin of the section)
Response: corrected
3. In line 224-The sentence “This agrees with the clustering of populations based on microsatellites (7)” in my opinion should be placed in Discussion section.
Response: Corrected – placed in Discussion section.